

# Screening of 49 antibiotic residues in aquatic products using modified QuEChERS sample preparation and UPLC-QToFMS analysis

Yao Gao[1], Tianwen Zhang[2], Shirong Huang[1], Xinxin Lin[1], Sisi Gong[1], Qiuhua Chen[2], Dongren Huang[2] and Min Chen[1]

[1] The School of Medical Technology and Engineering, Fujian Medical University, Fuzhou, China
[2] Fujian Fishery Resources Monitoring Center, Fuzhou, China

Corresponding authors
Yao Gao, yaogao@fjmu.edu.cn
Min Chen, cmjy503@163.com

## ABSTRACT

A precise analytical method was established for rapid screening of 49 antibiotic residues in aquatic products by ultra-high performance liquid chromatography-quadrupole time of flight mass spectrometry (UPLC-QToFMS). The quick, easy, cheap, effective, rugged and safe (QuEChERS) process was refined for effective sample preparation. The homogenized samples of aquatic products were extracted with 3% acetic acid in acetonitrile, salted out with anhydrous magnesium sulfate and sodium chloride, and cleaned up by octadecylsilane (C18) and primary-secondary amine (PSA) powder. Then, the purified samples were separated on a BEH C18 column using 0.1% formic acid and methanol as mobile phases by gradient elution, detected by MS under positive Electron Spray Ionization (ESI+) mode. The linear range of matrix-matched calibration curve was 1–100 µg/L for each compound with the correlation coefficients in the range of 0.9851–0.9999. The recoveries of target antibiotics at the different spiked levels ranged from 60.2% to 117.9% except for lincomycin hydrochloride, whereas relative standard deviations (RSDs) were between 1.6% and 14.0% except for sulfaguanidine in grass Carp, Penaeus vannamei and *Scylla serrata* matrices. The limits of detection (LODs) (S/N = 3) for the analytes were 0.05–2.40 µg/kg, 0.08–2.00 µg/kg and 0.10–2.27 µg/kg and the limits of quantification (LOQs) (S/N = 10) were 0.16–8.00 µg/kg, 0.25–6.66 µg/kg and 0.32–7.56 µg/kg in grass Carp, Penaeus vannamei and *Scylla serrata*, respectively. The method was successfully applied to grass Carp, Penaeus vannamei and *Scylla serrata*, demonstrating its ability for the determination of multi-categories antibiotic residues in aquatic products.

## INTRODUCTION

Antibiotics, as a vital medicine with bactericidal or bacteriostatic effect, are widely used in modern aquaculture to prevent infectious diseases and promote growth for the increase of aquatic production (*Liu et al., 2018*; *Liu, Steele & Meng, 2017*). However, antibiotics would be a dietary risk in cultured aquatic products with abuse of antibiotics happened.

Their residues may directly enter the human body and accumulate in human organs. Therefore, they could lead to a series of adverse reactions and toxicological effects, such as allergic reactions, toxic reactions, liver damage, kidney damage, nervous system damage, and so on (*Mo et al., 2017*). More seriously, the extensive usage of antibiotics could induce antimicrobial resistance which is considered as a public health threat (*Anderson et al., 2017*). Based on both major negative effects above, regulatory limits for veterinary medicine residues are worldwide issued by many countries and organizations like Ministry of Agriculture (MOA) of China No 235 and European Union (EU) No 37/2010 (*Delatour et al., 2018*). To protect consumers, the overall situation of antibiotic residues in aquatic products that serve as a main food source in coastal areas of China has gained increasing attention from governments.

At present, the analytical methods for antibiotics in animal food mainly include liquid chromatography (LC) (*Zhou et al., 2015*), liquid chromatography tandem triple quadrupole mass spectrometry (LC-MS/MS) (*Guidi et al., 2018*) and liquid chromatography hybrid quadrupole time-of-flight mass spectrometry (LC-QToFMS) (*Ki et al., 2019*). An LC method is always equipped with fluorescence detector which has the disadvantage of lower sensitivity and poorer qualitative ability. The major shortcoming of LC-MS/MS is a limited throughput when each compound needs optimization in instrumental parameter of mass spectrometer. With the significant advances in the performance of LC-QToFMS, this platform has the outstanding merits of high resolution, high sensitivity and applicability for high throughput screening analysis in aquatic products (*Gu et al., 2019*). Owing to its excellent characteristics, hereby an ultra performance LC-QToFMS (UPLC-QToFMS) was applied for the rapid determination of multi-categories antibiotic residues at levels below their general maximum residue limits (MRLs) (2–200μg/kg) as newly set by MOA (GB 31650-2019).

The quick, easy, cheap, effective, rugged and safe (QuEChERS) method introduced to improve extraction efficiency and to elevate method reliability in a great variety of samples, has been significantly developed and successfully applied in the residues analytical field (*Garcia & Gotah, 2017*; *Serra-Compte et al., 2017*). To our knowledge, previous researchers always focused on one sample type or a single class of veterinary drugs. *Villar-Pulido et al. (2011)* established a fast QuEChERS-LC-ToFMS method to detect 13 drug residues in shrimps. *Zhang et al. (2016)* used a QuEChERS procedure without solid-phase extraction step for rapid quantification of 90 kinds of veterinary drugs in royal jell. In this study, several kinds of aquatic products were continuously analyzed where efficiently extract multi-residues from the complex matrices is the most tough and trouble step. Therefore, development of a rapid, sensitive and simultaneous analytical method aiming at antibiotic residues at trace levels in aquatic products is urgent.

## MATERIALS AND METHODS

### Chemicals and solutions

A total of 49 antibiotics selected for the study contains four families including lincosamides (two), macrolides (nine), quinolones (16) and sulfonamides (22) (Table 1). Forty-nine

**Table 1 CAS number, molecular formula, molecular weight, RT, characteristic ions and structural formula of 49 antibiotics.**

| Antibiotic | CAS | Molecular formula | Molecular weight | RT (min) | Precursor ion (m/z) | Product ions (m/z) | Structural formula |
|---|---|---|---|---|---|---|---|
| Lincomycin hydrochloride | 859-18-7 | $C_{18}H_{35}ClN_2O_6S$ | 443.00 | 8.17 | 407.2213 | 126.1281,359.2176 | |
| Clindamycin hydrochloride | 21462-39-5 | $C_{18}H_{33}ClN_2O_5S$ | 461.44 | 11.76 | 425.1877 | 158.1179,590.3893 | |
| Azithromycin | 83905-01-5 | $C_{38}H_{72}N_2O_{12}$ | 748.99 | 10.86 | 749.5153 | 158.1180,591.4227 | |
| Leucomycin | 1392-21-8 | $C_{40}H_{67}NO_{14}$ | 785.96 | 13.19 | 786.4618 | 109.0657,174.1132, 558.3282 | |
| Clarithromycin | 81103-11-9 | $C_{38}H_{69}NO_{13}$ | 747.96 | 13.65 | 748.4853 | 158.1180,590.3899 | |
| Roxithromycin | 80214-83-1 | $C_{41}H_{76}N_2O_{15}$ | 837.05 | 13.77 | 837.5327 | 158.1185,679.4380 | |

(Continued)

| Table 1 (continued) | | | | | | | |
|---|---|---|---|---|---|---|---|
| Antibiotic | CAS | Molecular formula | Molecular weight | RT (min) | Precursor ion (m/z) | Product ions (m/z) | Structural formula |
| Tylosin | 1401-69-0 | $C_{46}H_{77}NO_{17}$ | 916.10 | 12.62 | 916.527 | 174.1131,772.4469 | |
| Erythromycin | 114-07-8 | $C_{37}H_{67}NO_{13}$ | 733.93 | 12.83 | 734.4663 | 158.1181,576.3743 | |
| Tilmicosin | 108050-54-0 | $C_{46}H_{80}N_2O_{13}$ | 869.15 | 11.43 | 869.5726 | 174.1134,696.4655 | |
| Spiramycin | 8025-81-8 | $C_{43}H_{74}N_2O_{14}$ | 843.06 | 10.46 | 843.5208 | 174.1128,540.3170 | |
| Virginiamycin M1 | 21411-53-0 | $C_{28}H_{35}N_3O_7$ | 525.59 | 13.34 | 526.2552 | 337.1193,508.2453 | |
| Enrofloxacin | 93106-60-6 | $C_{19}H_{22}FN_3O_3$ | 359.39 | 8.79 | 360.1717 | 245.1090,316.1823 | |

| Antibiotic | CAS | Molecular formula | Molecular weight | RT (min) | Precursor ion (m/z) | Product ions (m/z) | Structural formula |
|---|---|---|---|---|---|---|---|
| Norfloxacin | 70458-96-7 | $C_{16}H_{18}FN_3O_3$ | 319.33 | 8.54 | 320.1406 | 233.1084,276.1505 | |
| Pefloxacin | 70458-92-3 | $C_{17}H_{20}FN_3O_3$ | 333.35 | 8.37 | 334.156 | 233.1091,290.1666 | |
| Ciprofloxacin | 85721-33-1 | $C_{17}H_{18}FN_3O_3$ | 331.34 | 8.70 | 332.1404 | 314.1305, 231.0571, 288.1509 | |
| Ofloxacin | 82419-36-1 | $C_{18}H_{20}FN_3O_4$ | 361.37 | 8.36 | 362.1516 | 261.1043,318.1618 | |
| Sarafloxacin | 98105-99-8 | $C_{20}H_{17}F_2N_3O_3$ | 385.36 | 9.31 | 386.1315 | 299.0995, 342.1414, 368.1210 | |
| Enoxacin | 74011-58-8 | $C_{15}H_{17}FN_4O_3$ | 320.32 | 8.39 | 321.1377 | 232.0522,303.1255 | |
| Lomefloxacin | 98079-51-7 | $C_{17}H_{19}F_2N_3O_3$ | 351.35 | 8.99 | 352.1487 | 265.1143,308.1574 | |

(Continued)

| Table 1 (continued) | | | | | | | |
|---|---|---|---|---|---|---|---|
| Antibiotic | CAS | Molecular formula | Molecular weight | RT (min) | Precursor ion (m/z) | Product ions (m/z) | Structural formula |
| Nalidixic acid | 389-08-2 | $C_{12}H_{12}N_2O_3$ | 232.24 | 12.02 | 233.0928 | 187.0508,215.0816 | |
| Oxolinic acid | 14698-29-4 | $C_{13}H_{11}NO_5$ | 283.21 | 10.79 | 262.0717 | 244.0619 | |
| Flumequine | 42835-25-6 | $C_{14}H_{12}FNO_3$ | 261.25 | 12.32 | 262.0882 | 202.0298,244.0764 | |
| Danofloxacin | 112398-08-0 | $C_{19}H_{20}FN_3O_3$ | 357.38 | 8.82 | 358.1561 | 245.1083,340.1449 | |
| Difluoxacin hydrochloride | 91296-86-5 | $C_{21}H_{20}ClF_2N_3O_3$ | 435.85 | 9.11 | 400.1471 | 299.0991, 358.1569, 382.1362 | |
| Orbifloxacin | 113617-63-3 | $C_{19}H_{20}F_3N_3O_3$ | 395.38 | 9.06 | 396.1537 | 295.1054,352.1635 | |
| Sparfloxacin | 110871-86-8 | $C_{19}H_{22}F_2N_4O_3$ | 392.40 | 9.83 | 393.1739 | 292.1250,349.1827 | |

| Antibiotic | CAS | Molecular formula | Molecular weight | RT (min) | Precursor ion (m/z) | Product ions (m/z) | Structural formula |
|---|---|---|---|---|---|---|---|
| Fleroxacin | 79660-72-3 | $C_{17}H_{18}F_3N_3O_3$ | 369.34 | 8.10 | 370.1374 | 269.0893,326.1469 | |
| Sulfamerazine | 127-79-7 | $C_{11}H_{12}N_4O_2S$ | 264.30 | 7.30 | 265.0754 | 92.0496,156.0111 | |
| Sulfapyridine | 144-83-2 | $C_{11}H_{11}N_3O_2S$ | 249.29 | 6.90 | 250.0652 | 92.0495,156.0111 | |
| Sulfamethoxypyridazine | 80-35-3 | $C_{11}H_{12}N_4O_3S$ | 280.30 | 8.54 | 281.0703 | 92.0496, 126.0662, 156.0114 | |
| Sulfamethoxazole | 723-46-6 | $C_{10}H_{11}N_3O_3S$ | 253.28 | 9.05 | 254.0603 | 92.0497,156.0113 | |
| Sulfadoxine | 2447-57-6 | $C_{12}H_{14}N_4O_4S$ | 310.33 | 9.39 | 311.0817 | 92.0496,156.0115 | |
| Sulfathiazole | 72-14-0 | $C_9H_9N_3O_2S_2$ | 255.32 | 6.48 | 256.0212 | 92.0495,156.0111 | |
| sulfamethizole | 144-82-1 | $C_9H_{10}N_4O_2S_2$ | 270.33 | 8.20 | 271.0321 | 92.0495,156.0113 | |

(Continued)

| Antibiotic | CAS | Molecular formula | Molecular weight | RT (min) | Precursor ion (m/z) | Product ions (m/z) | Structural formula |
|---|---|---|---|---|---|---|---|
| Trimethoprim | 738-70-5 | $C_{14}H_{18}N_4O_3$ | 290.32 | 8.16 | 291.1467 | 123.0655, 261.0979, 275.1135 | |
| Sulfisoxazole | 127-69-5 | $C_{11}H_{13}N_3O_3S$ | 267.30 | 8.09 | 268.0757 | 92.0495,156.0112 | |
| Sulfamoxole | 729-99-7 | $C_{11}H_{13}N_3O_3S$ | 267.30 | 9.41 | 268.0756 | 92.0500, 113.0710, 156.0113 | |
| Sulfabenzamide | 127-71-9 | $C_{13}H_{12}N_2O_3S$ | 276.31 | 9.80 | 277.0643 | 92.0496,156.0113 | |
| Sulfaphenazole | 526-08-9 | $C_{15}H_{14}N_4O_2S$ | 314.36 | 10.13 | 315.0914 | 156.0111,158.0710 | |
| Sulfamethazine | 57-68-1 | $C_{12}H_{14}N_4O_2S$ | 278.33 | 8.30 | 279.0917 | 124.0828, 156.0119, 186.0330 | |
| Sulfadiazine | 68-35-9 | $C_{10}H_{10}N_4O_2S$ | 250.28 | 5.23 | 251.0596 | 92.0496,156.0112 | |
| Sulfaquinoxaline | 59-40-5 | $C_{14}H_{12}N_4O_2S$ | 300.34 | 10.81 | 301.076 | 146.0713,156.0114 | |
| Sulfachlorpyridazine | 80-32-0 | $C_{10}H_9ClN_4O_2S$ | 284.72 | 8.87 | 285.0206 | 92.0497,156.0115 | |
| Sulfameter | 651-06-9 | $C_{11}H_{12}N_4O_3S$ | 280.30 | 9.16 | 281.0701 | 92.0493, 126.0657, 156.0107 | |

| Antibiotic | CAS | Molecular formula | Molecular weight | RT (min) | Precursor ion (m/z) | Product ions (m/z) | Structural formula |
|---|---|---|---|---|---|---|---|
| Sulfisomidine | 515-64-0 | $C_{12}H_{14}N_4O_2S$ | 278.33 | 5.82 | 279.0917 | 124.0867,186.0328 | |
| Sulfamonomethoxine | 1220-83-3 | $C_{11}H_{12}N_4O_3S$ | 280.30 | 8.05 | 281.0706 | 126.0660,156.0111 | |
| Sulfadimethoxine | 122-11-2 | $C_{12}H_{14}N_4O_4S$ | 310.33 | 10.54 | 311.0817 | 92.0494,156.0764 | |
| Sulfaguanidine | 57-67-0 | $C_7H_{10}N_4O_2S$ | 214.24 | 1.89 | 215.0601 | 92.0494,156.0112 | |
| Sulfapyrazole | 852-19-7 | $C_{16}H_{16}N_4O_2S$ | 328.39 | 10.73 | 329.107 | 156.0121,172.0870 | |

antibiotic standards and six internal isotope standards (roxiyhromycin-D7, enrofloxacin-D5 hydrochloride, sulfadoxine-D3, ciprofloxacin-D8, norfloxacin-D5, and sulfadimethoxine-D6, purity:93.6%) were obtained from Dr. Ehrenstorfer GmbH (Germany). Methanol, acetonitrile, ethyl acetate were purchased from Merck (UPLC-grade; Darmstadt, Germany). Anhydrous sodium sulfate of analytical reagent grade and HPLC-grade formic acid, acetic acid, sodium chloride, octadecylsilane (C18), alumina-N (ALU-N), primary-secondary amine (PSA) and leucine enkephalin was provided by ANPEL (China).

Individual stock solutions (100 µg/mL) were prepared by dissolving each antibiotic standard in methanol and then stored at −18 °C. Mixed standard solution (1 µg/mL) were diluted from the stock solutions with methanol. Calibration curves were obtained by diluting mixed standard solution with acetonitrile—water solvent (25:75 v/v) at the final concentration of 1, 5, 10, 25, 50, 100 ng/mL. The concentrations of 6 isotope internal standards in each calibration standard solution were 20 ng/mL.

## Sample treatment

Three main species of aquatic products including grass Carp, Penaeus vannamei and *Scylla serrata*, which acted as common food in Fujian province were involved in this research.

After collection from supermarkets, 32 fresh samples of aquatic products were treated according to Practice of sampling plans for aquatic products (GB/T 30891-2014) including amount, size, transport and storage of sampling. To prevent antibiotic degradation, they were immediately stored in the refrigerator at −20 °C prior to analysis. Each kind of aquatic samples (2 ± 0.01 g) was thawed at room temperature and weighed into a 50 mL centrifuge tube. Afterwards, each tube was added with 50 μL mixed antibiotic standard solution (1 μg/mL) and then was mixed and placed for 15 min.

## Antibiotic extraction and clean-up optimization

The targeted residues were extracted using a modified QuEChERS method, which were optimized in terms of extractants, salting-out agents and sorbents. Antibiotics were extracted by 10 mL ACN with 3% acetic acid. Then, salting-out agent (3 g of anhydrous $Na_2SO_4$ and 1 g of NaCl) were successively placed into the tube and swirled for 1 min. Subsequently, the tube was centrifuged for 5 min at 10,000 rpm 4 °C. A 6.5 mL supernatant was transferred to a 15 mL centrifuge tube containing the sorbents of 200 mg C18 and 50 mg PSA. The tube was swirled for 2 min and then centrifuged for 10 min at 5,000 rpm 4 °C. Five milliliters aliquot of supernatant was pipetted to a 25-mL evaporation flask and dried using a rotary evaporator under a nitrogen flow at 50 °C. The residue was fully resuspended in 1 mL of acetonitrile-water solvent (25:75 v/v) by ultrasonication and oscillation. The solution was subsequently filtered through 0.22 μm nylon membrane before final placement into an auto-sampler vial for the UPLC-QToFMS analysis.

## Instrumental conditions

### Instrumental

ACQUITY H-CLASS UPLC and Xevo G2-S Q-ToF mass spectrometer (Waters, Milford, MA, USA) with electrospray ionization source were used. A 3–30K high speed refrigerated centrifuge (SiGMA, Ronkonkoma, NY, USA), MS3 digital vortex mixer (IKA, Königswinter, Germany), laborata 4000 efficient rotary evaporator (Heidolph, Schwabach, Germany), multi Reax oscillator (Heidolph, Schwabach, Germany), N-EVAP™ 112 (Organomation Associates, Berlin, MA, USA) and Milli-Q water purification system (Millipore, Burlington, MA, USA) were used for sample preparation.

### LC conditions

The separation of mixed antibiotic standard solutions were achieved on a Waters Acquity UPLC BEH C18 silica column (100 mm × 3.0 mm, 1.7 μm). A gradient LC elution method was employed by 0.1% formic acid aqueous solution as mobile phase A and methanol as mobile phase B.

The gradient elution was as follows: 10% B at 0–3 min, 10–100% B at 3–15 min, 100% B at 15–18 min, 100–10% B at 18–18.1 min and 10% B at 18.1–21 min. The injection volume, flow rate, sample manager and column temperature were set at 10 μL, 0.3 mL/min, 10 °C and 40 °C, respectively. All target antibiotics were eluted, and the column was cleaned and equilibrated.

### MS conditions

MS experiments were operated using electrospray ionization (ESI) in the positive mode. The optimum MS parameters were as follows: mass collection range 50–1,000 Da; capillary voltage 3.0 kV; ion source temperature 120 °C; desolvation temperature 450 °C; cone gas flow 50 L/h; desolvation gas flow rate 800 L/h and core voltage 40 V.

QToFMS screening for 49 antibiotic residues was performed using $MS^E$ mode. The simultaneous acquisition of accurate-mass full-spectrum at low and high collision energy are allowed in $MS^E$ mode, where the low collision energy (LE) spectrum provides useful information on the parent molecules and the main fragment ions were obtained commonly in the high collision energy (HE) function. In this study, LE was set as 6 V and HE was set from 10 eV to 40 eV. Leucine enkephalin, a commonly used peptide, was employed here as a reference material to tune MS instruments in every 10 s.

## RESULTS AND DISCUSSION

### Optimization of LC condition

The effect of the two types of mobile phases in the separation process were compared between 0.1% formic acid-acetonitrile and 0.1% formic acid water-methanol. As shown in Fig. 1, using 0.1% formic acid water-acetonitrile as the mobile phases, it is difficult to separate sulfamonomethoxine and sulfamethoxypyridazine completely. It was found that when methanol was used, better resolution and higher overall signal response were obtained. Therefore, 0.1% formic acid water-methanol was selected as the mobile phase in this experiment.

### Optimization of the QuEChERS process

#### Sample extraction

For the purpose of optimizing extraction of the antibiotic residues for different substrates of aquatic products including grass Carp, Penaeus vannamei and *Scylla serrata*, ethyl acetate and acetonitrile mixed with different amounts of acetic acid were compared. As shown in Fig.2, 3% acetic acid acetonitrile was used as the extractant, and the average recoveries of 49 antibiotics in three matrices were 75.3%, 76.7%, 81.8%, respectively, which were higher than using 1% acetic acid-acetonitrile (v:v), 5% acetic acid-acetonitrile (v:v), and ethyl acetate for the extraction. Intriguingly the acidity of the extractant has a great effect on the quinolones. The sequence of recoveries of quinolones from low to high was ethyl acetate, acetonitrile, 1% acetic acid acetonitrile, 3% acetic acid acetonitrile, 5% acetic acid acetonitrile when each of them was performed as the extractant.
The possible reason is that quinolones, which are amphoteric, are easily soluble in acidic or alkaline such as acetic acid solutions. From these results, 3% acetic acid acetonitrile was chosen as the optimum composition of solvents for the extraction buffer.

#### Purification procedure

Five most commonly used sorbents were investigated in this experiment, including PSA, C18, ALU-N, PSA-C18 mixture, PSA-ALU-N mixture. The purification effects on grass Carp, Penaeus vannamei and *Scylla serrata* were shown in Fig. 3. It is obvious that

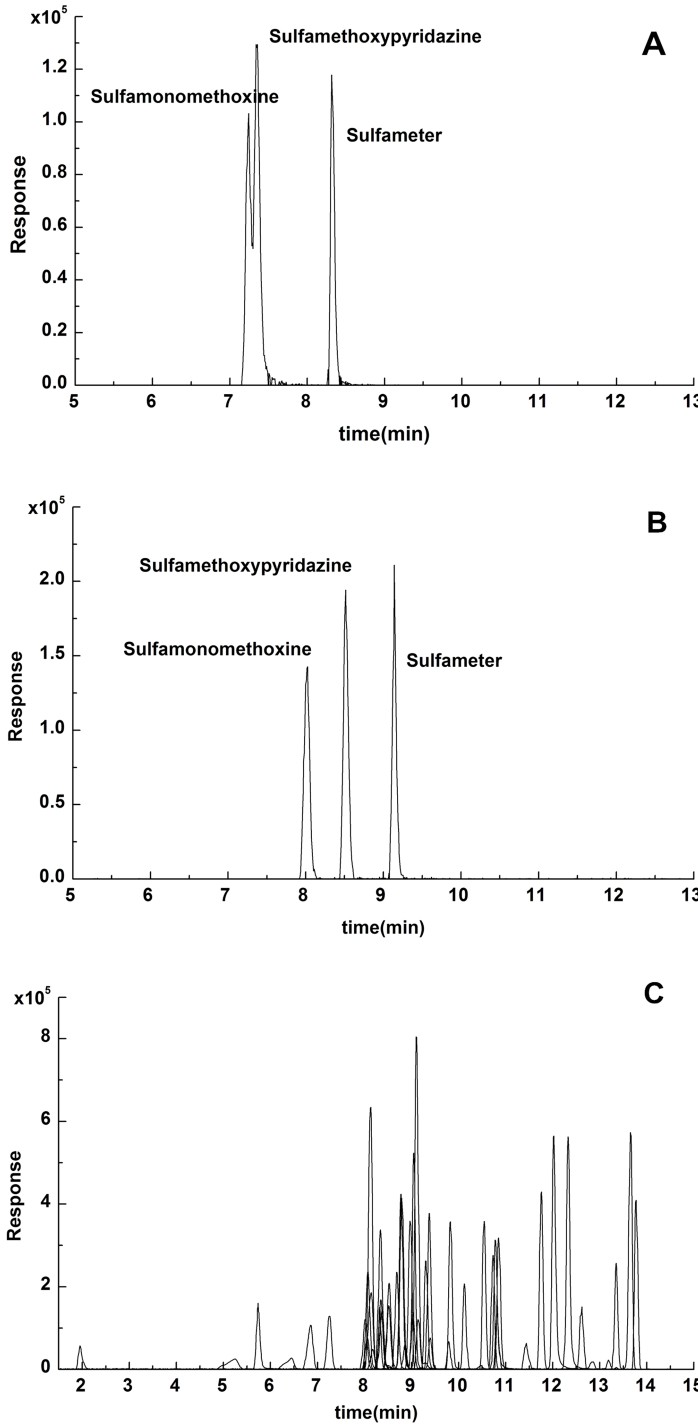

**Figure 1 Chromatogram of the three isomers of sulfamonomethoxine, sulfamethoxypyridazine and sulfameter with (A) 0.1% formic acid water-acetonitrile and (B) 0.1% formic acid water-methanol as the mobile phase, respectively. (C) Overlapping extracted ion chromatograms of 49 antibiotics with 0.1% formic acid water-methanol as the mobile phase.** Using 0.1% formic acid water-acetonitrile as the mobile phases, it is difficult to separate sulfamonomethoxine and sulfamethoxypyridazine completely.

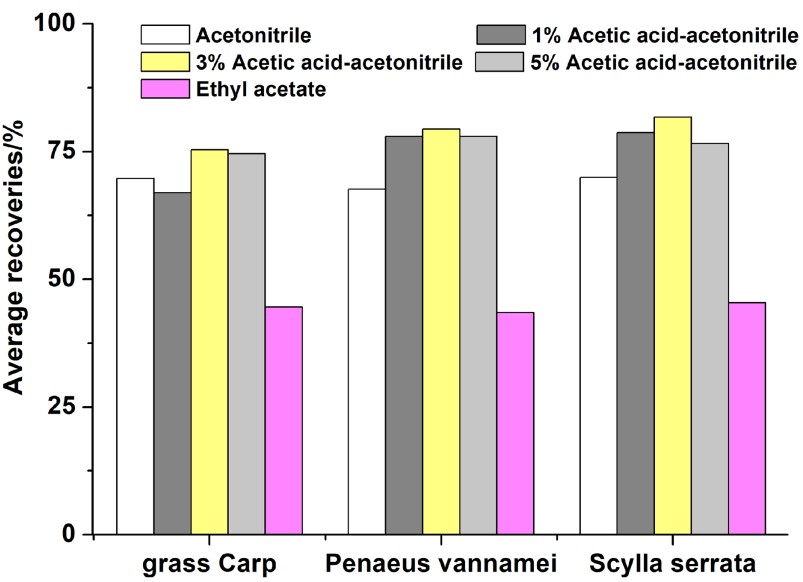

**Figure 2 Effects of different extracting solvents on the recoveries of the 49 antibiotics.** Using 3% acetic acid acetonitrile as the extractant, the average recoveries of 49 antibiotics in three matrices were 75.3%, 76.7%, 81.8%, respectively, which were higher than using 1% acetic acid-acetonitrile (v:v), 5% acetic acid-acetonitrile (v:v), and ethyl acetate for the extraction.

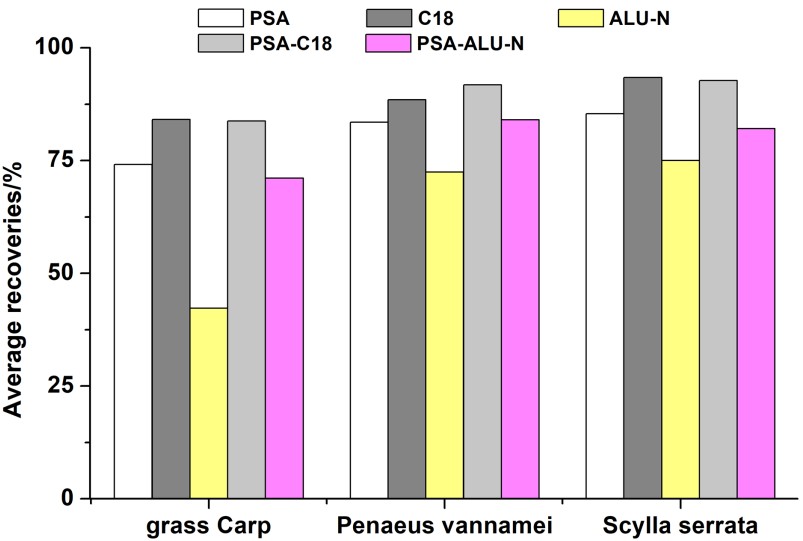

**Figure 3 Effects of five different sorbents on the average recoveries of the 49 antibiotics in grass Carp, Penaeus vannamei and *Scylla serrata*.** The highest average recoveries of all 49 antibiotics in three matrices were achieved using PSA-C18, overall.

ALU-N gets an inferior purification effect probably because ALU-N has a certain adsorption effect on antibiotics especially quinolones. The highest average recoveries of all 49 antibiotics in three matrices were achieved using PSA-C18, overall.

Afterwards, the amounts of salting-out agents (anhydrous $Na_2SO_4$ and NaCl) and sorbents (PSA and C18) were optimized using $L_9(3^4)$ orthogonal experimental design

**Table 2 Orthogonal design for sorbents and salting agents.**

| Levels | Factors | | |
|---|---|---|---|
| | PSA (mg) | C18 (mg) | $Na_2SO_4$: NaCl (g:g) |
| 1 | 50 | 100 | 4:1 |
| 2 | 100 | 200 | 3:1 |
| 3 | 150 | 300 | 2:1 |

at three levels (Table 2). The results indicated that satisfactory recoveries of
49 antibiotics were observed when 3g $Na_2SO_4$/1 g NaCl and 50 mg PSA/200 mg C18
were conducted.

After optimization, the average recoveries of 49 antibiotics in grass Carp, *Penaeus
vannamei* and *Scylla serrata* reached 83.4%, 88.4%, and 88.8% respectively, while this
procedure provided the best results for the majority of target antibiotics. In summary, this
improved QuEChERS process for antibiotic extraction in aquatic products is fast, effective,
economical and eco-friendly.

## Method validation
### Identification
As listed in Table 1, each of the 49 target antibiotics was measured in $MS^E$ mode by one
precursor ion and at least two product ions. Meanwhile, retention time was also required to
provide vital information to identify specific antibiotics.

### Linear range, regression equation, limits of detection and limits of quantitation
The series of solvent-based standard solutions were prepared according to "Chemicals and
Solutions" and were then determined by UPLC-QToFMS. The calibration curves were
obtained from the relationship between the analyte concentration (X, μg/L) and the
analyte peak areas/internal standard peak area, providing the linear equation and the
correlation coefficient for each analyte. The linear ranges were 1–100 μg/L for each
examined analyte with correlation coefficients of greater than 0.9888. The limits of
detection (LODs) were evaluated with signal-to-noise ratio (S/N) of 3 and the limits
of quantification (LOQs) were evaluated with S/N of 10. LODs and LOQs of solvent-
based calibration curves were in the range of 0.01–1.33 μg/L and 0.04–4.42 μg/L,
respectively.

### Matrix effects
Aquatic products are rich in proteins and unsaturated fatty acids, as well as they contain a
variety of vitamins, minerals, trace elements and so on. Complex components cause
ubiquitous matrix effects (signal suppression and enhancement) during the LC–MS/MS
analysis which may strongly affect the quantitative accuracy and reproducibility in this
study (*Guo et al., 2016*). Here, the matrix effects of three subtracts were evaluated by

**Table 3 Matrix effects, LODs and LOQs for all matrices tested.**

| Antibiotic | grass Carp | | Penaeus vannamei | | Scylla serrata | |
|---|---|---|---|---|---|---|
| | Matrix effect (%) | LOD/LOQ (μg/kg) | Matrix effect (%) | LOD/LOQ (μg/kg) | Matrix effect (%) | LOD/LOQ (μg/kg) |
| Lincomycin hydrochloride | 31.79 | 0.21/0.71 | 41.39 | 0.83/2.77 | 43.54 | 0.77/2.55 |
| Clindamycin hydrochloride | −25.94 | 0.31/1.04 | −1.62 | 0.28/0.94 | −24.43 | 0.31/1.03 |
| Azithromycin | −11.18 | 0.12/0.41 | −7.69 | 0.26/0.86 | 0.31 | 0.17/0.58 |
| Clarithromycin | 17.16 | 0.05/0.16 | 32.76 | 0.08/0.25 | 29.59 | 0.17/0.56 |
| Roxithromycin | −9.19 | 0.07/0.23 | −18.25 | 0.09/0.30 | −32.34 | 0.14/0.45 |
| Tylosin | 51.20 | 0.18/0.61 | 50.99 | 0.26/0.86 | 47.37 | 0.42/1.39 |
| Erythromycin | −8.76 | 2.40/8.00 | −1.99 | 1.12/3.73 | 7.00 | 1.78/5.93 |
| Tilmicosin | 27.82 | 0.36/1.18 | 15.82 | 0.48/1.59 | 36.00 | 0.66/2.20 |
| Spiramycin | −0.50 | 1.32/4.40 | −7.63 | 1.65/5.50 | −2.03 | 1.38/4.60 |
| Virginiamycin M1 | 28.51 | 0.48/1.60 | 31.29 | 0.39/1.29 | 42.49 | 0.24/0.81 |
| Enrofloxacin | 6.17 | 0.33/1.09 | 4.81 | 0.41/1.35 | 5.34 | 0.40/1.34 |
| Norfloxacin | 4.56 | 0.56/1.86 | 12.74 | 0.74/2.47 | 15.72 | 1.35/4.51 |
| Pefloxacin | 26.43 | 0.60/1.99 | 27.39 | 0.55/1.85 | 7.24 | 1.14/3.81 |
| Ciprofloxacin | −14.44 | 0.20/0.65 | −8.25 | 0.33/1.11 | −12.74 | 0.49/1.63 |
| Ofloxacin | −29.83 | 0.65/2.18 | −25.24 | 0.25/0.84 | −43.51 | 0.51/1.69 |
| Sarafloxacin | −5.64 | 0.38/1.27 | 5.55 | 0.15/0.49 | 7.29 | 0.42/1.40 |
| Enoxacin | 12.29 | 1.44/4.80 | 5.83 | 1.54/5.15 | 12.33 | 2.09/6.98 |
| Lomefloxacin | 3.40 | 0.29/0.98 | 5.78 | 0.26/0.85 | 15.48 | 0.61/2.04 |
| Nalidixic acid | −3.27 | 0.26/0.88 | 8.22 | 0.22/0.75 | 0.68 | 0.19/0.62 |
| Oxolinic acid | −10.55 | 0.18/0.60 | 2.98 | 0.38/1.26 | −4.17 | 0.56/1.88 |
| Flumequine | −15.30 | 0.22/0.74 | 5.03 | 0.15/0.51 | −24.12 | 0.33/1.09 |
| Danofloxacin | −5.79 | 0.20/0.68 | −7.85 | 0.66/2.20 | −0.75 | 0.65/2.15 |
| Difluoxacin hydrochloride | −17.38 | 0.16/0.53 | −5.70 | 0.08/0.28 | −3.54 | 0.13/0.45 |
| Orbifloxacin | 4.17 | 0.13/0.43 | 4.59 | 0.11/0.36 | −0.25 | 0.16/0.53 |
| Sparfloxacin | −5.40 | 0.23/0.77 | −21.83 | 0.20/0.65 | −35.49 | 0.34/1.13 |
| Fleroxacin | 3.25 | 0.31/1.03 | −14.68 | 0.80/2.65 | −29.59 | 0.69/2.31 |
| Sulfamerazine | 2.09 | 0.29/0.98 | 30.66 | 0.17/0.57 | 18.46 | 0.23/0.78 |
| Sulfapyridine | 1.84 | 0.23/0.77 | 12.30 | 0.30/0.99 | −10.27 | 0.24/0.80 |
| Sulfamethoxypyridazine | −12.69 | 0.55/1.83 | 0.76 | 0.58/1.95 | 28.26 | 0.10/0.34 |
| Sulfamethoxazole | 1.90 | 0.12/0.41 | 10.38 | 0.27/0.89 | 4.80 | 0.45/1.50 |
| Sulfadoxine | −3.65 | 0.21/0.69 | 9.84 | 0.19/0.63 | 0.73 | 0.12/0.40 |
| Sulfathiazole | 6.85 | 0.24/0.79 | 13.20 | 0.52/1.73 | 13.02 | 0.15/0.49 |
| Sulfamethizole | −5.47 | 0.60/2.01 | 7.40 | 0.72/2.41 | 3.91 | 0.40/1.32 |
| Trimethoprim | 0.25 | 0.10/0.34 | −1.28 | 0.08/0.26 | −1.78 | 0.10/0.32 |
| Sulfisoxazole | −11.43 | 0.21/0.69 | 3.77 | 0.20/0.66 | 15.40 | 1.18/3.94 |
| Sulfamoxole | −19.74 | 0.37/1.23 | −24.75 | 0.50/1.66 | −26.04 | 0.20/0.67 |
| Sulfabenzamide | −11.43 | 0.42/1.41 | −0.98 | 0.83/2.76 | 0.55 | 1.00/3.34 |
| Sulfaphenazole | 1.42 | 0.29/0.97 | 27.97 | 0.35/1.17 | 29.30 | 0.80/2.67 |
| Sulfamethazine | −5.14 | 2.13/7.11 | 17.09 | 2.00/6.66 | 21.71 | 2.04/6.79 |

*(Continued)*

| Table 3 (continued) | | | | | | |
|---|---|---|---|---|---|---|
| Antibiotic | grass Carp | | Penaeus vannamei | | Scylla serrata | |
| | Matrix effect (%) | LOD/LOQ (µg/kg) | Matrix effect (%) | LOD/LOQ (µg/kg) | Matrix effect (%) | LOD/LOQ (µg/kg) |
| Sulfadiazine | −9.30 | 0.60/2.02 | 1.00 | 0.42/1.39 | 28.47 | 0.76/2.53 |
| Sulfaquinoxaline | −22.11 | 0.37/1.25 | −16.77 | 0.42/1.39 | −2.79 | 0.89/2.96 |
| Sulfachlorpyridazine | −3.70 | 0.25/0.82 | 18.58 | 0.60/1.99 | 19.14 | 0.36/1.20 |
| Sulfameter | −17.09 | 0.64/2.15 | −4.26 | 0.48/1.60 | 0.49 | 0.67/2.25 |
| Sulfisomidine | −24.53 | 1.86/6.20 | −21.60 | 1.89/6.31 | −13.95 | 2.27/7.56 |
| Sulfamonomethoxine | −9.24 | 0.28/0.92 | −24.85 | 0.35/1.18 | −38.07 | 0.66/2.20 |
| Sulfadimethoxine | −5.97 | 0.34/1.13 | −6.84 | 0.24/0.81 | −7.14 | 0.20/0.66 |
| Sulfaguanidine | −13.11 | 1.54/5.14 | −12.19 | 1.82/6.07 | −10.28 | 2.00/6.67 |
| Sulfapyrazole | −15.58 | 0.15/0.5 | −17.05 | 0.22/0.72 | −19.00 | 0.16/0.52 |

comparing the calibration curves of the target antibiotics prepared in solvent and in the matrix (Hernando et al., 2007), which is calculated as:

$$\text{Matrix effect } (\%) = (\text{Slope}_{\text{matrix-matched standard curve}}/\text{Slope}_{\text{solvent-based standard curve}} - 1) \times 100$$

Three sets of blank matrix samples were introduced to the mixed standard solution of different concentrations (1, 5, 10, 25, 50, 100 µg/L). As listed in Table 3, among the three matrices of grass Carp, Penaeus vannamei and Scylla serrata, matrix effects could still encountered in determining several antibiotics such as lincomycin hydrochloride, clindamycin hydrochloride and tylosin. Therefore, matrix-matched standard curves were applied to mitigate matrix effects for quantification of 49 antibiotics. The results of the regression analysis showed that the correlation coefficients ($R^2$) of the matrix-matched standard curves of 49 antibiotics in grass Carp, Penaeus vannamei and Scylla serrata ranged from 0.9900 to 0.9999, 0.9851 to 0.9998, 0.9908 to 0.9997, respectively which indicated excellent linearity.

Based on data obtained from matrix-matched standard curves of 49 antibiotics in grass Carp, Penaeus vannamei and Scylla serrata, the range of the LODs were 0.05–2.40 µg/kg, 0.08–2.00 µg/kg and 0.10–2.27 µg/kg, respectively. And LOQs were in the range of 0.16–8.00 µg/kg, 0.25–6.66 µg/kg and 0.32–7.56 µg/kg, respectively. Hereby, the results of all the LODs and LOQs exhibited in Table 3 in this research were satisfactory as compared with the MRLs.

### Recovery and precision

In order to investigate the accuracy and precision of this method, recovery experiments were conducted at different spiking levels of 10, 50, 100 µg/kg (Table 4). Among the 49 antibiotics, except for lincomycin hydrochloride whose recoveries were less than 60%, the recoveries of other antibiotics in three matrices were generally greater than 70%. These results indicated that this method had a satisfactory stability and could meet the actual detecting requirements of 49 antibiotics in aquatic products.

**Table 4 Recoveries and repeatability (expressed as %RSD) results for all matrices tested.**

| Antibiotic | Spiked levels (μg/kg) | grass Carp | | Penaeus vannamei | | Scylla serrata | |
|---|---|---|---|---|---|---|---|
| | | Recovery/% | RSD/% | Recovery/% | RSD/% | Recovery/% | RSD/% |
| Lincomycin hydrochloride | 10 | 54.1 | 5.4 | 37.9 | 6.7 | 44.0 | 10.4 |
| | 50 | 55.5 | 2.7 | 37.4 | 4.8 | 32.2 | 11.6 |
| | 100 | 50.7 | 3.1 | 39.6 | 5.4 | 39.3 | 15.7 |
| Clindamycin hydrochloride | 10 | 76.4 | 10.9 | 76.6 | 5.5 | 81.4 | 4.2 |
| | 50 | 73.8 | 5.7 | 76.0 | 5.3 | 73.0 | 11.5 |
| | 100 | 74.9 | 4.1 | 100.5 | 3.5 | 82.8 | 6.4 |
| Azithromycin | 10 | 100.0 | 10.4 | 104.8 | 6.0 | 111.2 | 3.7 |
| | 50 | 81.8 | 5.4 | 101.6 | 4.6 | 95.6 | 5.5 |
| | 100 | 100.2 | 7.7 | 116.0 | 2.0 | 104.3 | 3.1 |
| Leucomycin | 10 | 81.2 | 7.0 | 86.8 | 4.9 | 63.8 | 3.7 |
| | 50 | 82.8 | 7.4 | 88.7 | 5.7 | 69.4 | 3.5 |
| | 100 | 73.4 | 8.0 | 93.4 | 7.8 | 77.9 | 5.6 |
| Clarithromycin | 10 | 89.8 | 7.3 | 95.8 | 5.7 | 98.6 | 5.3 |
| | 50 | 96.6 | 4.4 | 102.0 | 2.4 | 95.9 | 6.4 |
| | 100 | 88.7 | 3.8 | 100.4 | 4.7 | 105.1 | 2.2 |
| Roxithromycin | 10 | 91.0 | 2.2 | 94.8 | 2.2 | 89.1 | 5.0 |
| | 50 | 79.8 | 3.6 | 87.4 | 5.0 | 73.5 | 6.3 |
| | 100 | 84.6 | 3.8 | 90.3 | 2.6 | 83.1 | 6.0 |
| Tylosin | 10 | 77.3 | 7.7 | 87.6 | 6.1 | 104.7 | 5.4 |
| | 50 | 76.8 | 4.6 | 91.4 | 5.1 | 99.2 | 3.7 |
| | 100 | 74.1 | 3.6 | 103.3 | 3.2 | 101.3 | 6.4 |
| Erythromycin | 10 | 88.3 | 9.1 | 97.8 | 14.0 | 93.1 | 5.5 |
| | 50 | 76.1 | 4.7 | 78.0 | 8.3 | 75.7 | 5.1 |
| | 100 | 78.0 | 3.1 | 66.6 | 5.5 | 64.8 | 5.0 |
| Tilmicosin | 10 | 93.9 | 7.3 | 97.9 | 6.9 | 89.1 | 6.5 |
| | 50 | 80.8 | 3.4 | 95.3 | 4.8 | 100.7 | 3.1 |
| | 100 | 97.2 | 7.2 | 101.4 | 3.5 | 106.4 | 2.8 |
| Spiramycin | 10 | 74.7 | 11.3 | 91.7 | 8.6 | 100.7 | 4.8 |
| | 50 | 60.2 | 10.8 | 74.7 | 5.1 | 73.1 | 3.4 |
| | 100 | 64.6 | 4.5 | 85.9 | 11.1 | 71.9 | 5.5 |
| Virginiamycin M1 | 10 | 73.0 | 12.7 | 102.4 | 4.1 | 103.4 | 4.9 |
| | 50 | 75.7 | 6.6 | 98.2 | 3.4 | 88.7 | 8.9 |
| | 100 | 68.1 | 6.0 | 107.4 | 4.8 | 91.0 | 4.6 |
| Enrofloxacin | 10 | 99.2 | 4.9 | 109.1 | 2.6 | 101.4 | 4.4 |
| | 50 | 90.4 | 2.4 | 107.1 | 3.0 | 100.4 | 2.8 |
| | 100 | 95.6 | 4.7 | 104.5 | 3.1 | 101.8 | 1.7 |
| Norfloxacin | 10 | 104.0 | 4.3 | 84.4 | 6.0 | 87.6 | 4.5 |
| | 50 | 101.2 | 6.9 | 84.4 | 4.0 | 90.1 | 7.1 |
| | 100 | 103.6 | 5.3 | 87.1 | 4.4 | 93.3 | 5.6 |

*(Continued)*

| | Table 4 (continued) | | | | | | |
|---|---|---|---|---|---|---|---|
| Antibiotic | Spiked levels (µg/kg) | grass Carp | | Penaeus vannamei | | Scylla serrata | |
| | | Recovery/% | RSD/% | Recovery/% | RSD/% | Recovery/% | RSD/% |
| Pefloxacin | 10 | 86.1 | 5.3 | 105.1 | 8.1 | 108.4 | 4.2 |
| | 50 | 96.4 | 9.3 | 106.2 | 4.0 | 108.2 | 4.1 |
| | 100 | 101.0 | 5.8 | 104.1 | 4.1 | 105.1 | 2.7 |
| Ciprofloxacin | 10 | 78.9 | 2.8 | 86.0 | 2.4 | 94.3 | 5.5 |
| | 50 | 86.0 | 3.8 | 83.8 | 3.9 | 98.7 | 4.9 |
| | 100 | 91.7 | 4.5 | 90.5 | 5.7 | 103.1 | 5.4 |
| Ofloxacin | 10 | 83.8 | 4.0 | 101.8 | 6.6 | 95.3 | 6.8 |
| | 50 | 97.5 | 5.2 | 106.3 | 2.4 | 109.0 | 4.2 |
| | 100 | 92.4 | 5.5 | 97.4 | 4.8 | 105.1 | 3.7 |
| Sarafloxacin | 10 | 85.7 | 6.3 | 81.3 | 3.8 | 86.6 | 5.6 |
| | 50 | 91.7 | 3.7 | 85.5 | 5.1 | 94.4 | 5.4 |
| | 100 | 97.0 | 4.7 | 100.2 | 7.3 | 98.0 | 8.1 |
| Enoxacin | 10 | 92.3 | 6.8 | 96.6 | 9.7 | 89.0 | 5.5 |
| | 50 | 95.9 | 7.1 | 103.1 | 2.9 | 105.6 | 4.0 |
| | 100 | 98.0 | 5.6 | 100.4 | 2.6 | 104.0 | 5.3 |
| Lomefloxacin | 10 | 92.2 | 7.1 | 88.0 | 4.5 | 102.3 | 6.1 |
| | 50 | 89.4 | 8.6 | 84.7 | 4.5 | 102.8 | 4.3 |
| | 100 | 104.3 | 4.9 | 98.3 | 5.4 | 104.1 | 2.7 |
| Nalidixic acid | 10 | 77.8 | 6.6 | 74.3 | 7.5 | 66.7 | 4.0 |
| | 50 | 103.3 | 4.0 | 87.5 | 4.0 | 75.8 | 3.9 |
| | 100 | 106.5 | 2.7 | 97.4 | 2.6 | 82.4 | 3.6 |
| Oxolinic acid | 10 | 78.2 | 8.6 | 70.3 | 5.5 | 65.8 | 3.8 |
| | 50 | 99.2 | 11.0 | 86.4 | 5.0 | 76.8 | 4.2 |
| | 100 | 102.3 | 3.2 | 94.2 | 3.9 | 81.3 | 2.3 |
| Flumequine | 10 | 75.0 | 8.9 | 74.2 | 8.9 | 66.8 | 3.2 |
| | 50 | 106.2 | 5.1 | 88.7 | 3.6 | 80.9 | 4.4 |
| | 100 | 103.3 | 2.0 | 95.8 | 4.3 | 85.8 | 2.1 |
| Danofloxacin | 10 | 112.9 | 2.0 | 117.9 | 2.8 | 105.2 | 4.2 |
| | 50 | 95.6 | 2.7 | 101.5 | 4.2 | 105.2 | 2.4 |
| | 100 | 100.7 | 4.2 | 102.1 | 3.5 | 106.5 | 3.5 |
| Difluoxacin hydrochloride | 10 | 92.0 | 2.8 | 79.8 | 6.1 | 95.0 | 7.2 |
| | 50 | 94.8 | 3.5 | 79.8 | 6.1 | 104.6 | 3.3 |
| | 100 | 102.4 | 2.0 | 98.3 | 4.9 | 104.5 | 2.5 |
| Orbifloxacin | 10 | 72.8 | 11.5 | 74.0 | 3.8 | 79.3 | 4.3 |
| | 50 | 92.9 | 8.6 | 85.6 | 5.5 | 97.8 | 4.1 |
| | 100 | 99.9 | 3.9 | 96.7 | 3.0 | 102.7 | 3.2 |
| Sparfloxacin | 10 | 75.1 | 5.0 | 75.7 | 6.8 | 63.6 | 3.2 |
| | 50 | 79.8 | 4.5 | 93.5 | 3.1 | 88.9 | 5.6 |
| | 100 | 78.0 | 1.7 | 106.0 | 4.1 | 100.6 | 3.8 |

| Table 4 (continued) | | | | | | | |
| --- | --- | --- | --- | --- | --- | --- | --- |
| **Antibiotic** | **Spiked levels (μg/kg)** | **grass Carp** | | **Penaeus vannamei** | | *Scylla serrata* | |
| | | **Recovery/%** | **RSD/%** | **Recovery/%** | **RSD/%** | **Recovery/%** | **RSD/%** |
| Fleroxacin | 10 | 116.0 | 6.0 | 110.7 | 7.6 | 108.1 | 3.1 |
| | 50 | 111.9 | 4.2 | 105.5 | 5.4 | 104.4 | 3.4 |
| | 100 | 102.2 | 5.0 | 106.3 | 2.2 | 104.2 | 4.6 |
| Sulfamerazine | 10 | 67.1 | 6.4 | 86.3 | 5.8 | 90.6 | 2.6 |
| | 50 | 80.4 | 6.4 | 81.6 | 2.1 | 75.2 | 7.0 |
| | 100 | 72.7 | 3.0 | 81.4 | 6.6 | 71.6 | 4.0 |
| Sulfapyridine | 10 | 71.6 | 3.6 | 88.4 | 8.4 | 88.1 | 4.9 |
| | 50 | 78.4 | 8.5 | 76.7 | 2.7 | 82.2 | 6.1 |
| | 100 | 79.5 | 3.8 | 76.1 | 6.7 | 86.4 | 3.6 |
| Sulfamethoxypyridazine | 10 | 74.7 | 5.1 | 88.0 | 5.0 | 87.1 | 2.4 |
| | 50 | 73.9 | 5.1 | 75.2 | 2.6 | 76.4 | 2.8 |
| | 100 | 77.9 | 7.7 | 84.1 | 2.1 | 80.8 | 3.0 |
| Sulfamethoxazole | 10 | 73.7 | 4.2 | 85.9 | 9.5 | 82.7 | 3.5 |
| | 50 | 74.7 | 7.4 | 82.9 | 1.9 | 71.0 | 7.4 |
| | 100 | 76.4 | 3.6 | 78.8 | 6.0 | 69.9 | 3.9 |
| Sulfadoxine | 10 | 69.1 | 4.1 | 88.4 | 4.5 | 84.8 | 3.3 |
| | 50 | 75.2 | 4.0 | 82.0 | 5.3 | 71.0 | 4.3 |
| | 100 | 77.1 | 3.0 | 83.4 | 2.8 | 72.7 | 3.3 |
| Sulfathiazole | 10 | 72.3 | 3.4 | 88.6 | 10.1 | 73.1 | 8.8 |
| | 50 | 73.0 | 8.5 | 90.6 | 4.9 | 85.4 | 6.0 |
| | 100 | 71.8 | 2.4 | 86.4 | 4.6 | 90.7 | 4.2 |
| Sulfamethizole | 10 | 65.6 | 4.7 | 95.6 | 3.9 | 86.7 | 6.2 |
| | 50 | 72.6 | 5.3 | 75.2 | 3.8 | 85.6 | 6.3 |
| | 100 | 72.2 | 2.8 | 85.4 | 4.2 | 92.8 | 7.3 |
| Trimethoprim | 10 | 85.3 | 8.6 | 92.1 | 4.0 | 85.3 | 5.1 |
| | 50 | 105.6 | 5.1 | 95.2 | 3.0 | 90.8 | 4.3 |
| | 100 | 100.4 | 5.4 | 94.7 | 3.6 | 103.2 | 1.6 |
| Sulfisoxazole | 10 | 71.6 | 3.3 | 85.3 | 9.6 | 81.9 | 5.3 |
| | 50 | 78.7 | 7.4 | 74.3 | 5.1 | 85.9 | 5.3 |
| | 100 | 74.4 | 2.9 | 91.9 | 4.9 | 103.4 | 3.3 |
| Sulfamoxole | 10 | 76.9 | 3.5 | 94.3 | 3.8 | 89.0 | 4.6 |
| | 50 | 78.3 | 5.0 | 85.2 | 4.5 | 81.6 | 4.8 |
| | 100 | 78.4 | 3.4 | 91.1 | 3.5 | 87.0 | 3.7 |
| Sulfabenzamide | 10 | 81.6 | 3.2 | 94.8 | 5.6 | 86.9 | 4.6 |
| | 50 | 83.3 | 5.8 | 85.6 | 5.0 | 84.9 | 6.8 |
| | 100 | 76.9 | 6.0 | 94.4 | 2.1 | 90.6 | 6.1 |
| Sulfaphenazole | 10 | 83.8 | 2.4 | 90.6 | 3.1 | 84.7 | 3.9 |
| | 50 | 97.0 | 8.5 | 81.4 | 4.6 | 75.1 | 5.3 |
| | 100 | 84.1 | 2.6 | 86.7 | 5.7 | 73.3 | 2.9 |

(Continued)

| Table 4 (continued) | | grass Carp | | Penaeus vannamei | | Scylla serrata | |
|---|---|---|---|---|---|---|---|
| Antibiotic | Spiked levels (µg/kg) | Recovery/% | RSD/% | Recovery/% | RSD/% | Recovery/% | RSD/% |
| Sulfamethazine | 10 | 79.1 | 3.2 | 85.6 | 11.4 | 87.0 | 3.9 |
| | 50 | 79.6 | 5.9 | 76.8 | 2.0 | 74.6 | 4.8 |
| | 100 | 75.0 | 3.4 | 78.9 | 3.2 | 72.0 | 3.6 |
| Sulfadiazine | 10 | 80.3 | 4.3 | 92.6 | 7.1 | 90.4 | 4.9 |
| | 50 | 83.2 | 7.7 | 89.4 | 5.6 | 83.0 | 3.4 |
| | 100 | 79.1 | 2.7 | 86.7 | 5.2 | 82.5 | 3.1 |
| Sulfaquinoxaline | 10 | 80.3 | 4.3 | 85.0 | 3.9 | 88.9 | 3.9 |
| | 50 | 83.2 | 7.6 | 78.7 | 2.6 | 74.9 | 4.7 |
| | 100 | 79.1 | 2.7 | 86.2 | 1.6 | 75.7 | 5.4 |
| Sulfachlorpyridazine | 10 | 78.3 | 3.3 | 87.8 | 6.6 | 86.1 | 5.6 |
| | 50 | 72.9 | 3.3 | 83.6 | 2.7 | 74.8 | 4.7 |
| | 100 | 69.8 | 3.7 | 82.6 | 4.4 | 73.3 | 3.8 |
| Sulfameter | 10 | 82.0 | 4.4 | 90.9 | 3.7 | 83.1 | 2.7 |
| | 50 | 80.1 | 8.3 | 81.8 | 2.4 | 90.4 | 6.9 |
| | 100 | 75.5 | 5.3 | 89.6 | 2.9 | 90.3 | 7.6 |
| Sulfisomidine | 10 | 75.3 | 2.7 | 89.2 | 4.8 | 86.3 | 4.0 |
| | 50 | 74.8 | 4.4 | 81.9 | 4.6 | 86.7 | 6.5 |
| | 100 | 74.6 | 3.1 | 84.4 | 3.0 | 90.4 | 6.9 |
| Sulfamonomethoxine | 10 | 78.1 | 3.0 | 90.1 | 3.9 | 87.8 | 7.9 |
| | 50 | 77.2 | 7.2 | 94.0 | 4.9 | 96.9 | 4.5 |
| | 100 | 76.6 | 3.6 | 96.1 | 4.2 | 105.1 | 2.6 |
| Sulfadimethoxine | 10 | 79.8 | 6.7 | 91.7 | 3.9 | 91.9 | 2.3 |
| | 50 | 71.6 | 8.2 | 83.4 | 3.7 | 83.1 | 6.9 |
| | 100 | 74.7 | 5.1 | 88.7 | 4.3 | 92.4 | 6.5 |
| Sulfaguanidine | 10 | 75.8 | 6.0 | 68.2 | 9.9 | 91.8 | 4.3 |
| | 50 | 85.1 | 7.4 | 58.8 | 34.1 | 77.8 | 5.5 |
| | 100 | 77.3 | 4.8 | 64.8 | 5.0 | 67.4 | 8.9 |
| Sulfapyrazole | 10 | 81.3 | 2.1 | 93.3 | 3.2 | 82.1 | 4.1 |
| | 50 | 95.6 | 5.6 | 80.1 | 4.5 | 77.5 | 6.2 |
| | 100 | 85.7 | 2.7 | 80.7 | 2.7 | 84.9 | 5.8 |

## Application to real samples

In this study, 32 samples of aquatic products (including 12 grass Carp, 11 Penaeus vannamei, and 9 *Scylla serrata*) bought from supermarkets were tested to display the applicability of this method. These samples were dealt with the improved QuEChERS procedure and screened by UPLC-QToFMS. All antibiotic residues were quantified using the matrix-matched calibration method, increasing the data accuracy. Results showed that difluoxacin hydrochloride was detected in the samples of Penaeus vannamei whose amounts ranged from 1.5 to 7.0 µg/kg. MRLs of difluoxacin hydrochloride was 300 µg/kg according to GB 31650-2019 announced by MOA, China. Overall, all the concentrations of

antibiotic residues in real samples were lower than their MRLs, while other target antibiotics were below their LOQs.

## CONCLUSIONS

Summing up, in this study, a fast, convenient, effective, economical and eco-friendly strategy based on QuEChERS process was established to extract the antibiotics in aquatic products including grass Carp, Penaeus vannamei and *Scylla serrata*. Using UPLC-QToFMS platform and matrix-matched calibration method to screen and quantity the 49 antibiotic residues, the study achieved satisfactory recoveries, significant linearity and decent stability. Our method also possesses great potential in the analysis of various kinds of antibiotic residues in aquatic products.

### Funding

This research was financially supported by National Natural Science Foundation of China (Grant No. 21405017) and the Natural Science Foundation of Fujian Province (Grant No. 2018J01676). The funders had no role in study design, data collection and analysis, decision to publish, or preparation of the manuscript.

### Grant Disclosures

The following grant information was disclosed by the authors:
National Natural Science Foundation of China: 21405017.
Natural Science Foundation of Fujian Province: 2018J01676.

### Competing Interests

The authors declare that they have no competing interests.

### Author Contributions

- Yao Gao conceived and designed the experiments, performed the experiments, analyzed the data, performed the computation work, prepared figures and/or tables, authored or reviewed drafts of the paper, and approved the final draft.
- Tianwen Zhang conceived and designed the experiments, performed the experiments, analyzed the data, performed the computation work, prepared figures and/or tables, and approved the final draft.
- Shirong Huang performed the experiments, analyzed the data, performed the computation work, prepared figures and/or tables, authored or reviewed drafts of the paper, and approved the final draft.
- Xinxin Lin analyzed the data, prepared figures and/or tables, and approved the final draft.
- Sisi Gong performed the experiments, performed the computation work, authored or reviewed drafts of the paper, and approved the final draft.
- Qiuhua Chen performed the experiments, analyzed the data, performed the computation work, prepared figures and/or tables, and approved the final draft.

- Dongren Huang conceived and designed the experiments, authored or reviewed drafts of the paper, and approved the final draft.
- Min Chen conceived and designed the experiments, authored or reviewed drafts of the paper, and approved the final draft.

## Data Availability

The raw measurements are available as a Supplemental File.

## Supplemental Information

Supplemental information for this article can be found online at http://dx.doi.org/10.7717/peerj-achem.8#supplemental-information.

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
