# Peer review of "Screening of 49 antibiotic residues in aquatic products using modified QuEChERS sample preparation and UPLC-QToFMS analysis"

_PeerJ Analytical Chemistry, doi:10.7717/peerj-achem.8_

## Round 0.1 · original submission · Major Revisions

Please address all the comments of the reviewers carefully.

Reviewer 1 ·

Basic reporting

1. The English language should be improved, such as line 49, lines 120-121, lines 129-130;
2. The format of references should meet the requirements of the Journal;

Experimental design

No comment

Validity of the findings

1. Line 20, RSDs were between 1.6% and 14.0% in different samples were inconsistent with the datas given in Table 4, should check it;
2. Figure 1a-1c, the title of the Y-axis were unclear;
3, form Figure 3, the effects of C18 and PSA-C18 nearly the same, even in grass Carp and Scylla serrata higher average recoveries were achieved using C18, why select PSA-C18 as the sorbents?
4, In Table 1, the structure of target compounds was unclear.

Additional comments

The reviewed manuscript describes the development and application of a new analytical method for determining antibiotic residues in aquatic products by an optimised QuEChERS method and UPLC-QToFMS determination. The information in systematically well described and organised. The design of experiments for the QuEChERS and cleanup steps is well organized. The interest of the manuscript is high and it could be published in PeerJ after minor modifications.

Reviewer 2 ·

Basic reporting

The figure quality should be improved, the reference format should be consistent with the journal.

Experimental design

1. May be modified as “Screening of 49 antibiotic residues in aquatic products using modified QuEChERS sample preparation procedure and UPLC-QToFMS analysis
2. Abstract, Line 19 to 20, The recoveries of target antibiotics at the different spiked levels ranged from 60.6% to 117.9% …, 60.6% should be changed to 60.2% from table 4.
3. Introduction, a small description of regulatory limits of pesticide residues in China/EU/Codex/USA may be given to justify the LOQ and LOD requirement of the method.
4. Materials and methods, a separate section may be given for a brief description on different treatments used with respect to each optimization parameters may be given for the clarity of readers.

Validity of the findings

1. The spiked concentration selected is 10 μg/L, 50 μg/L, and 100 μg/L, however, as the LOQ of the most of the pesticides are 1.0 μg/L, first level recovery should be of 1.0 μg/L. Further ensure that, the highest calibration level should be 200 μg/L or more.
2. In a multi-residue method, the crux of the method is the optimization of sample preparation, especially in a matrix like aquatic products. Hence, more emphasis should be given for sample preparation optimization, matrix effect minimization, clean-up optimization, sample size optimization, sample preprocessing optimization and further validation.

Additional comments

The authors presented a good piece of work on analysis of multi-residue antibiotics in aquatic products. The manuscripts used sound analytical technology (UPLC-QToFMS) for antibiotic residue analysis. However, some of the gaps detailed below should be addressed carefully before publishing the paper in PeerJ.

---

## Round 0.2 · accepted · Accept

The authors have addressed the reviewers' comments.